# A Few-Shot Learning Approach Assists in the Prognosis Prediction of Magnetic Resonance-Guided Focused Ultrasound for the Local Control of Bone Metastatic Lesions

**DOI:** 10.3390/cancers14020445

**Published:** 2022-01-17

**Authors:** Fang-Chi Hsu, Hsin-Lun Lee, Yin-Ju Chen, Yao-An Shen, Yi-Chieh Tsai, Meng-Huang Wu, Chia-Chun Kuo, Long-Sheng Lu, Shauh-Der Yeh, Wen-Sheng Huang, Chia-Ning Shen, Jeng-Fong Chiou

**Affiliations:** 1The Ph.D. Program for Translational Medicine, College of Medical Science and Technology, Taipei Medical University and Academia Sinica, Taipei 110, Taiwan; d622103003@tmu.edu.tw; 2Department of Radiology, School of Medicine, College of Medicine, Taipei Medical University, Taipei 110, Taiwan; b001089024@tmu.edu.tw; 3Department of Radiation Oncology, Taipei Medical University Hospital, Taipei 110, Taiwan; b8601093@tmu.edu.tw (C.-C.K.); lslu@tmu.edu.tw (L.-S.L.); 4Genomics Research Center, Academia Sinica, Taipei 115, Taiwan; 5Graduate Institute of Biomedical Materials and Tissue Engineering, College of Biomedical Engineering, Taipei Medical University, Taipei 110, Taiwan; yjchen1113@tmu.edu.tw; 6International Ph.D. Program in Biomedical Engineering, College of Biomedical Engineering, Taipei Medical University, Taipei 110, Taiwan; 7TMU Research Center of Cancer Translational Medicine, Taipei Medical University, Taipei 110, Taiwan; 8Department of Medical Research, Taipei Medical University Hospital, Taipei 110, Taiwan; 9Department of Pathology, School of Medicine, College of Medicine, Taipei Medical University, Taipei 110, Taiwan; shen1202@tmu.edu.tw; 10Graduate Institute of Clinical Medicine, College of Medicine, Taipei Medical University, Taipei 110, Taiwan; 11International Master/Ph.D. Program in Medicine, College of Medicine, Taipei Medical University, Taipei 110, Taiwan; 12Department of Radiation Oncology, Shuang Ho Hospital, Taipei Medical University, Taipei 110, Taiwan; yijack@gmail.com; 13Department of Orthopedics, Taipei Medical University Hospital, Taipei 110, Taiwan; maxwutmu@tmu.edu.tw; 14Department of Orthopaedics, School of Medicine, College of Medicine, Taipei Medical University, Taipei 110, Taiwan; 15Ph.D. Program for Cancer Molecular Biology and Drug Discovery, College of Medical Science and Technology, Taipei Medical University and Academia Sinica, Taipei 110, Taiwan; 16School of Health Care Administration, College of Management, Taipei Medical University, Taipei 110, Taiwan; 17Department of Radiation Oncology, Wanfang Hospital, Taipei Medical University, Taipei 110, Taiwan; 18International Ph.D. Program for Cell Therapy and Regenerative Medicine, College of Medicine, Taipei Medical University, Taipei 110, Taiwan; 19Department of Urology, School of Medicine, College of Medicine, Taipei Medical University, Taipei 110, Taiwan; yehsd@tmu.edu.tw; 20Department of Urology, Taipei Medical University Hospital, Taipei 110, Taiwan; 21Cancer Center, Taipei Medical University Hospital, Taipei 110, Taiwan; 22Department of Nuclear Medicine, Taipei Medical University Hospital, Taipei 110, Taiwan; 201008@h.tmu.edu.tw

**Keywords:** magnetic resonance-guided focused ultrasound surgery, machine learning, prognosis prediction, bone metastasis, HIFU

## Abstract

**Simple Summary:**

We report a local control prediction model for patients undergoing MRgFUS ablation, and provide promising guidance for clinicians to identify a suitable treatment strategy for bone metastatic lesions. We propose a few-shot learning approach to establish the quick prediction of clinical and radiographic responses. On the basis of demographic data, pre-/post-treatment immune-related cytokine change, and MRI imaging, the most suitable parameters were selected to assess potential treatment outcomes during the acute inflammatory stages within 24 h. Traditional logistic regression and few-shot learning models were compared to identify the best model on an independent test. The best predictive few-shot learning model (accuracy of 85.2%, sensitivity of 88.6%, and AUC of 0.95) was achieved by combining the clinical features with the levels of significant cytokines IL-6, IL-13, IP-10, and eotaxin.

**Abstract:**

Magnetic resonance-guided focused ultrasound surgery (MRgFUS) constitutes a noninvasive treatment strategy to ablate deep-seated bone metastases. However, limited evidence suggests that, although cytokines are influenced by thermal necrosis, there is still no cytokine threshold for clinical responses. A prediction model to approximate the postablation immune status on the basis of circulating cytokine activation is thus needed. IL-6 and IP-10, which are proinflammatory cytokines, decreased significantly during the acute phase. Wound-healing cytokines such as VEGF and PDGF increased after ablation, but the increase was not statistically significant. In this phase, IL-6, IL-13, IP-10, and eotaxin expression levels diminished the ongoing inflammatory progression in the treated sites. These cytokine changes also correlated with the response rate of primary tumor control after acute periods. The few-shot learning algorithm was applied to test the correlation between cytokine levels and local control (*p* = 0.036). The best-fitted model included IL-6, IL-13, IP-10, and eotaxin as cytokine parameters from the few-shot selection, and had an accuracy of 85.2%, sensitivity of 88.6%, and AUC of 0.95. The acceptable usage of this model may help predict the acute-phase prognosis of a patient with painful bone metastasis who underwent local MRgFUS. The application of machine learning in bone metastasis is equivalent or better than the current logistic regression.

## 1. Introduction

Metastasis is a major complication that causes unrelieved pain and reduced quality of life in patients with cancer with advanced-stage malignancies [1,2]. The bone is a major metastatic site in the various cancers such as breast, lung, and prostate cancers. In many cases, the bone as the major metastatic site reflects high incidence rates, such as 65–75% in breast cancer, 65–75% in prostate cancer, and 30–40% in lung cancer, and the clinical significance of cancer recurrence [3,4]. Approximately 30–75% of patients in advanced stages suffer from bone metastasis and its related events, including bone pain and pathological fractures, particularly in weight-bearing bones [5,6]. Moreover, bone metastasis consistently affects the patients’ quality of life, performance status, and independent functioning. A previous study reported that >50% of patients who had received palliative pain controls for their painful bone metastatic lesions experienced improvements in their prognosis and quality of life [7]. Therefore, a suitable local control against bone metastatic lesions can significantly improve the prognosis and quality of life of patients in advanced stages.

The palliative radiotherapy of bone metastasis is a conventional clinical intervention that acts as an excellent option for long-term pain management, and pain relief can be achieved in 60–80% of patients, with 27% of patients experiencing pain recurrence during the first 3 months after treatment [8,9]. According to previous studies, magnetic resonance-guided focused ultrasound surgery (MRgFUS), which is used for conformal heat ablation, demonstrated rapid pain palliation within 1 month and was used against bone metastasis without radiation dose accumulation [10,11,12]. Therefore, MRgFUS is a relatively flexible treatment option for patients with different types of bone lesion, including osteolytic or osteoblastic, and pain recurrence [13]. In addition, MRgFUS is not limited by the maximal organ dose of radiotherapy, and is a palliative treatment option with radiotherapy and chemotherapy [14,15,16]. MRgFUS serves as a first-line treatment strategy for painful bone metastasis in selected patients.

On the basis of the advantages of the high acoustic absorption of cortical bone, MRgFUS could achieved more specific energy delivery in bone lesions, which provides high resolution compared with soft tissue [17,18,19]. High-intensity focused ultrasound (HIFU) conducts acoustic energy into deep-seated targets and achieves the local control of bone metastatic lesions. Previous studies suggested that thermal ablation might activate systemic antitumor immunity and result in long-term protection [20,21]. However, most patients with bone metastasis have end-stage disease and are expected to have poor prognosis. Suitable prognostic and predictive factors for the long-term outcomes after MRgFUS are still limited [22]. Further understanding posttreatment immunity and cytokine changes is in the spotlight of thermal medicine research [23]. After MRgFUS treatment, acute inflammatory cytokines such as interleukin (IL)-1β, IL-6, IL-8, IL-18, and tumor necrosis factor-α (TNF-α) are released into plasma [24,25,26]. The levels of IL-10 and transforming growth factor-β (TGF-β) also increase during the acute phase [24]. In addition to some proimmunity cytokines such as IL-1β and IL-8, IL-6, IL-10, and TGF-β may suppress anticancer immunity. The tendency of cytokine changes after thermal ablation may be fundamental in determining the prognosis of patients with bone metastasis.

Machine-learning (ML) algorithms support clinical decision making by efficiently classifying patients into different subtypes [27,28]. Biomarkers can also be identified on the basis of parameter selection via ML approaches. Evaluations of cytokine concentrations showed relationships to prognosis after MRgFUS treatment. However, the complexity of cytokine combinations is a bottleneck of traditional methods to elucidate the local control rate. Such multidimensional analysis requires more samples to achieve statistical significance [29,30,31]. The few-shot learning approach was used for high-throughput omic profiling to interpret genomewide data in a small-scale study population [30]. The features of this approach help in training the algorithm by using a small sample size of patients undergoing MRgFUS treatment. 

This study analyzes cytokine levels from the serum samples of 20 patients with bone metastatic lesions before and after MRgFUS thermal ablation. We examined 27 serum cytokine and chemokine levels before and after MRgFUS treatment within 24 h. The few-shot ML algorithm was used to classify the clinical parameters and cytokines. This few-shot learning approach assists in determining the clinical-response-related cytokine set that is applied into a prognosis determination model. The performance of logistic regression and few-shot learning models was compared to identify the most suitable prediction model.

## 2. Materials and Methods

### 2.1. Patient Eligibility

This study was approved by the Taipei Medical University-Joint Institutional Review Board (TMU-JIRB, approval number: N201803094). All patients who had undergone MRgFUS for bone metastasis between 1 December 2014 and 30 November 2017 at the Taipei Medical University Hospital, Taiwan were recruited after obtaining their informed consent. MRgFUS palliative treatment was suggested to patients who had at least one distinguishable painful bone metastasis with a numerical rating scale (NRS) pain score of >4. Patients kept original courses of systemic treatments, such as chemotherapy, bone-targeted agent, and pain management during the periods of 2 weeks before and 3 months after MRgFUS. MRgFUS can access targeted lesions without impending pathological fracture [10,22]. 

In this study, we retrospectively selected patients who had undergone palliative MRgFUS treatment. All participants fit the following inclusion criteria of our few-shot modeling. (1) Initial pain score with a numerical rating scale (NRS) score of >4. (2) MRgFUS was the first local therapy at target bone metastatic lesion without prior local radiotherapy. (3) Patient with Karnofsky performance status (KPS) of ≥60. (4) Patient with blood samples before MRgFUS treatment and after MRgFUS treatment within 24 h. (5) Patient obtained complete medical records, such as medication, survival, and regular image follow-up of 3 months after the MRgFUS. 

Exclusion criteria were local therapy before or within 3 months after MRgFUS, changing pain medication during the follow-up period, and the presence of pathological fractures in the lesions. Blood samples from the selected cases were collected in 10 mL EDTA tubes before and within 24 h after MRgFUS. Each batch of blood samples was stored in the Joint Biobank, Taipei Medical University (TMU-JBB). 

### 2.2. MRgFUS Treatment

The standard MRgFUS palliative treatment includes pretreatment computed tomographic scan, and obtaining T1- and T2-weighted sequences through variable orientations with and without spectral fat suppression using a 1.5 Tesla MR scanner (Signa HDxt, GE Healthcare, Waukesha, WI, USA) at the Taipei Medical University Hospital [22]. Pretreatment imaging provided the baseline images to identify the regions of treatment, skin contours, bone cortex, and nonpenetrating areas using the Exablate 2000 system (InSightec Ltd., Haifa, Israel). The Exablate system designs a treatment plan on the basis of the physician’s contouring of target volume marks, and HIFU was delivered to target the bone cortex with real-time MRI monitoring. Sonication energy from an ultrasound transducer pinpointed several heating regions to cover the entire target volume. The postablation coverage was confirmed using contrast-enhanced T1-weighted sequences.

### 2.3. Cytokine Analysis

The levels of 27 cytokines and chemokines were measured in the serum samples obtained from all participants; analysis was performed using the Bio-Plex Pro Human Cytokine 27-plex Assay (Bio-Rad Laboratories, Hercules, CA, USA). The following cytokines were assessed: IL-1β, IL-1ra, IL-2, IL-4, IL-6, IL-7, IL-8, IL-9, IL-10, IL-12p70, IL-13, IL-15, and IL-17; basic fibroblast growth factor (FGF); eotaxin; granulocyte-colony stimulating factor; granulocyte macrophage-colony stimulating factor; interferon (IFN)-γ; IFN-γ-induced protein-10 (IP-10); monocyte chemoattractant protein-1; macrophage inflammatory protein (MIP)-1α; MIP-1β; platelet-derived growth factor (PDGF)-BB; regulated upon activation, normal T-cell-expressed and secreted; TNF-α; and vascular endothelial growth factor (VEGF). Samples were preprocessed according to the manufacturer’s instructions. A gradient centrifuge with Ficoll-Paque™ PREMIUM (GE Healthcare Bio-sciences AB, Uppsala, Sweden) was used to separate plasma from whole-blood samples. Assay standard dilutions and 50 μL of plasma were added into a 96-well plate containing magnetic beads. After incubation for 30 min and then washing, streptavidin-conjugated phycoerythrin was added into the plate before final incubation. We used the Bio-Plex Array Reader system 2200 (Bio-Rad) to record data from the 96-well plate. To avoid the batch effect, samples from the same subjects were measured at the same time.

### 2.4. Few-Shot Learning and Predictive Model Construction

Supervised machine-learning (ML) models assist in model construction to detect cancer prognosis, including the random forest, generalized linear, support vector machine, and naive Bayesian models [32]. Few-shot learning was more suitable to help classify high-dimensional cytokine data from a small sample size (*n* = 20). Two prognostic methods, central neural network and gated recurrent unit, were then developed within this architecture. A point-biserial correlation test was used to evaluate the correlation of cytokine levels with the clinical response to MRgFUS treatment. All inputs of parameters included not only demographic data such as sex and age, but also clinical characteristics such as primary site, treatment site, coverage, minimal disease activity radiographic responses, and clinical responses, as shown in Figure 1B. Parameters were selected following the few-shot learning approach in the training dataset, and all parameters that attained statistical significance were included in the prediction model. The accuracy of prediction was compared between the few-shot and stepwise logistic regression models. We also set a confusion matrix to evaluate the performance of the prediction models. A prediction accuracy threshold was assessed using the confusion matrix, for which a success rate of prediction >60% was considered acceptable.

The prognostic model was established on the basis of the training dataset including 75% of the overall data that included all randomly selected samples (*n* = 15) and a validation dataset that included all patients (*n* = 20). Logistic regression analysis was performed to identify the most significant clinical factors from the training dataset, remove factors with less significant parameters from univariate analysis, and validate the predictive model with all covariables included in multivariate analysis. The odds ratio (OR) and the corresponding 95% confidence interval (CI) of parameters were estimated from the generalized linear model, logistic regression with stepwise selection. Collinearity between each independent variable and excluded variables was determined by the threshold variance inflation factor (VIF) > 4. Variables that yielded statistically significant results during univariate analysis of logistic regression (*p* < 0.05) were considered candidates for the further multivariate logistic regression analysis. Akaike information criterion (AIC) analysis was used to compare the goodness of the models. A receiver operating characteristic curve and Veall–Zimmermann’s Pseudo-R2 were calculated to suggest the optimized model, and Youden’s index was used to access the best cutoff value for relative cytokine levels after MRgFUS treatment. 

### 2.5. Model Comparison and Diagnosis

We followed different approaches and levels of model complexity (e.g., generalized linear model, logistic regression, and few-shot learning). We selected parameters from both forward and backward stepwise logistic regression. Parameters of generalized linear model (GLM), logistic regression, and a self-designed few-shot learning model were selected by statistically significant *p* and beta values. The performance of classification models was determined by Akaike information criteria (AIC) and the area under the curve (AUC). Models with an AUC above 80% were included in the model comparison. 

### 2.6. Statistical Analysis

Statistical analysis was performed using statistical software mlr package with R version 4.1.1 and GraphPad Prism 7 (San Diego, CA, USA) [33]. Cytokine concertation, KPS score, and NRS score were considered as continuous variables, and the means of two independent groups were compared by using the paired *t*-test and the Mann–Whitney U test. Clinical response, stage, and other categorical variables were tested using the χ^2^ test. All performed analyses were two-sided, and the threshold for statistical significance was determined as two-tailed *p* values < 0.05.

## 3. Results

### 3.1. Patient Demographics

A total of 20 patients with bone metastatic lesions were enrolled in this study from December 2014 to November 2017. Patient demographics are detailed in Table 1, and an overview of the study is presented in Figure 1A. From these patients, 40 peripheral venous blood samples were collected before and within 24 h after MRgFUS. 

In our registry, pretreatment demographics were listed as follows. The mean age was 63.95 ± 10.57 (range, 36–78) years, which was similar to the findings of our previous studies [10,22]. There were 8 males (40%) and 12 females (60%). Quality of life was presented in terms of the mean KPS ± 7.33 (range, 70–90); the baseline mean pain score was 6.65 ± 1.73 (range, 4–9). According to our pretreatment clinical assessments, 9 patients (45%) experienced moderate pain (NRS pain score 4–6), and 11 (55%) experienced severe pain (NRS pain score 7–10). Treatment sites covered almost all common targets in the sacroiliac joint (6 patients, 30%) and sacrum (5 patients, 25%). 

### 3.2. Cytokine Changes and Clinical Response

Cytokine levels were assessed for all recruited patients after MRgFUS treatment within 24 h compared with the pretreatment levels. Clinical responses were recorded at three months after treatment at least. A total of 16 patients demonstrated a complete response (CR) within 3 months of follow-up, and 4 patients exhibited a partial response with an 80% overall clinical response rate. Radiographic evaluation also revealed a response rate similar to the clinical response rate (overall radiographic response rate: 67.7%). During the acute phase of thermal ablation, we compared plasma cytokine levels between pre- and post-treatment phases (Table 2). IL-6 and IP-10 decreased significantly in the first 24 h after thermal ablation (IL-6, *p*-value: 0.049; IP-10, *p*-value: 0.04). IL-13 (*p*-value: 0.075) and eotaxin (*p*-value: 0.067) were also close to the significance threshold. These cytokine responses demonstrated an anti-inflammation tendency with fewer side effects in clinical follow-ups, such as fever, local swelling, and widespread dermatitis. Most other proinflammatory cytokines such IL-1β, TNF-α, VEGF, and PDGF and immune-regulatory cytokines also increased after the acute phase; however, this increase was not significant.

### 3.3. Treatmet Parameters

The MRgFUS treatment plan was calculated with the planning software of the Exablate 2000 system (InSightec, Haifa, Israel) before treatment. The sonication process was executed by following the treatment plans to cover the whole target volume. All treatment parameters, such as temperature, sonication time, and average acoustic power, were monitored with real-time temperature monitoring using proton resonance frequency (PRF) MR imaging. In this study, all 20 treatments of selected cases obtained 21.42 points of sonication. The delivered acoustic power was 51.08 ± 19.97 W, and the mean energy applied were 1029.93 ± 360.46 J. All sonication points were heated at 67.71 ± 7.36 °C in average (Table 3). 

### 3.4. Prediction Model of Clinical Response

All clinical parameters and cytokine profiles were selected through both multivariate analysis and few-shot learning, and the prediction model evaluated the prediction accuracy of the CR rates for patients who had undergone MRgFUS for bone metastatic lesions. Univariate analysis revealed several parameters associated with clinical response, including KPS (*p* = 0.04), IL-6 (*p* = 0.04), and IP-10 (*p* = 0.07), and the *p* values attained the threshold for statistical significance (Table 3). All clinical parameters and cytokine levels were selected through stepwise approaches applied to multivariate analysis for elucidating the correlation between the clinical responses of local controls. KPS (OR = 1.23, 95% CI = 1.00–1.74, and *p* = 0.04), age (OR = 1.02, 95% CI = 0.83–1.24, and *p* = 0.81), and IL-6 (OR = 0.77, 95% CI = 0.48–1.08, and *p* = 0.16) were also selected, and these maintained the correlation with the clinical responses (Table 4). The predictive accuracy of the final multivariate model with logistic regression parameters regarding clinical responses demonstrated significant accuracy (accuracy = 88% and chi-squared *p* value = 0.02; Table 5). Moreover, the few-shot learning approach included more cytokine parameters to predict the clinical response status, and revealed further accuracy and model-fitting results on the basis of the confusion matrix (accuracy = 95% and chi-squared *p* value = 0.002; Table 5). Area under the curve (AUC) values were also used to compare predictive abilities between the two different models: logistic regression and few-shot learning. The predictive model demonstrated high accuracy and reliability in terms of AUC and AIC scores. The logistic regression model showed an AUC of 0.88 and AIC of 19.35, whereas the few-shot learning model showed an AUC of 0.95 and an AIC of 17.14 (Figure 2).

## 4. Discussion

Few-shot learning is a specific approach that provides eigenvalue extraction and classification within small training datasets [29,31,32,34]. This technique aims to establish a classifier to recognize unseen features (target domain) from existing models (source domain). In this study, a novel few-shot learning prognostic model was proposed for palliative MRgFUS treatment with fewer patients than those in conventional radiotherapy. This algorithm embedded a deep-learning framework that had been trained by training samples, and model parameters were selected by stepwise logistic regression. Features of the validation dataset (target domain) were identified through cross-domain calculation, and accuracy was predicted by a confusion matrix. Parameters of the final model were compared regarding significance in each proposed model, and fine-tuned on the target models for final model optimization. The performance of the final model demonstrated a feasible accuracy of response prediction by using 20 cases with bone metastasis. 

This study analyzed the predictive and prognostic factors for the long-term outcomes of patients who underwent MRgFUS as a primary pain palliation therapy for bone metastases. Higher pretreatment KPS correlated with greater pain reduction, which remained consistent with our previous results [22] and demonstrated suitable lesion control with radiographic scoring criteria. After 3 months of follow-up, 80% of patients showed CRs without any major complications. A previous matched-pair study emphasized that MRgFUS shows rapid effects in the first month of follow-up, and palliative radiotherapy attains the complete local control in the third month [10]. Furthermore, the comparison of the efficacy between MRgFUS and bone radiotherapy may reflect a better long-term local control rate than that of conventional bone radiotherapy. MRgFUS achieved pain relief at ablation sites within 3 days. Approximately 72% of the patients maintained their CR status after 3 months [10]. This finding opens a window for selecting MRgFUS as an efficient intervention against bone metastasis. Despite MRgFUS showing promising pain-control ability, the use of thermal ablation against solid tumors and bone metastatic lesions remains controversial. The main concern associated with thermal ablation is the paucity of evidence regarding local recurrence, and wound healing leads to improvements in the residual tumor after the thermal ablation [35,36]. The long-term local control of thermal ablation involves the modulation of the immune microenvironment into an antitumor environment by combining anticancer drugs with tumor ablation. 

In this study, proinflammatory cytokines such as IL-6 and IP-10 demonstrated decreasing trends. IL-6 and IP-10 were both selected and incorporated into the prediction models obtained through stepwise multiple regression analysis and few-shot learning. The significant decrease reflects the creation of an anti-inflammatory microenvironment via thermal ablation. Such an anti-inflammatory environment normally leads to wound healing and cell proliferation by triggering the VEGF, PDGF, and FGF pathways [37,38]. However, IL-6, as a key factor of the cytokine-driven IL-13/JAK/STAT3 pathway during tumorigenesis, decreased in the acute inflammation phase after MRgFUS [37]. The activation of STAT3 in a tumorigenic region improves cancer cell proliferation and inhibits apoptosis, which may be the reason for in situ tumor recurrence after thermal ablation. On the other hand, IL-1β may exert protumor effects through myeloid-derived suppressor cells to release IL-10 and downregulate IL-12 secretion from macrophages. As a double-edged sword, IL-1β also stimulates CD8+ T cells and generates antitumor potential. As per our understanding, the crucial factor that determines pro- or antitumorigenesis is the dynamic combination of cytokines. The level of a single cytokine is insufficient to predict the pattern of tumor recurrence. 

The prediction model of the few-shot machine-learning technique was evaluated by parameter selection and model performance. The prediction power was identified by a confusion matrix that only allowed for the model to achieve over 70% of successful predictions from each permutation test. The area under the ROC curve (AUC) as the performance metric for analysis was applied for model comparison. Maximizing the AUC of the final model indicated 95% accuracy. As a prognostic model that assists in decision making, this selected few-shot model provided sophisticated sensitivity and specificity for local control rate. However, this study is still limited by the case number and follow-up time to achieve more accurate estimation for different patients. In order to better estimate long-term pain control and survival status, time-dependent cytokine concentration monitoring and imaging evaluation could further elucidate the interactions of cytokines and prognosis. 

The results of logistic regression analysis (Table 5) showed that KPS scores, patient age, and IL-6 and IP-10 levels correlate with the clinical response after MRgFUS tumor ablation. Accuracy and AUC show the reliability of the conventional logistic regression model. Nevertheless, few-shot learning suggested more parameters such as TNF-α and INF-γ and their interactions to construct a more comprehensive model. The performance of this few-shot learning model was more precise for prognosis prediction. The few-shot learning approach classified the potential parameters involved in the prognosis and clinical response. However, the exact mechanism and essential cytokines remain unclear. Further validation is warranted to identify the potential parameters in other datasets. These metalearning designs may optimize the prediction models of MRgFUS in local bone metastasis control.

In order to overcome the limitation of the few patients, we suggest a novel application of few-shot learning algorithm for predict the prognosis after MRgFUS palliative treatment. The few-shot learning approach was specified to extract eigenvalues and classify hidden subsets [31,32,34,39]. The algorithm embedded a Bayesian framework that minimized training samples by prior parameter selections. Our sensitivity analysis demonstrated that the expansion of datasets facilitated the accuracy and robustness of the prediction model (Appendix A and Appendix A). K-fold cross validation showed that the AUCs of prediction model were within an acceptable level from 0.86 to 0.97 when the dataset was splitting into different subsets by various folds (k = 1, 3, and 10).

Clinical outcomes of local control in bone metastasis were indicated by patient-reported NRS score and pain medication, which may have been affected by individual difference. Imaging evaluation for treated bone lesion provides a more accurate determination of the local control responder. The radiographic response as a stringent evaluation tool was applied to properly validate prognosis and clinical outcome. MDA criteria that are used to distinguish the therapeutic response in bone metastases were modified for single-site bone metastatic lesions by contrast-enhanced CT or MRI before and 3 months after treatment [22,40]. Overall, 16 responders (11 CR cases and 5 PR cases) in our dataset were identified as the radiographic response rate of 80% (16 responders/20 patients) with the MDA criteria. The performance of the radiographic final model showed accuracy of 91% and area under the curve of ROC of 0.89 (Appendix A and Appendix A). The AIC with radiographic response was 108.65 and AIC with clinical response was 8.85. Comparing the two criteria, the few-shot learning model showed feasible performance by using both MDA criteria (imaging assessment) and clinical indices for prognosis predictions.

In summary, this few-shot learning model generated a pipeline to classify meaningful cytokine parameters and clinical indices for prognosis after palliative MRgFUS treatments. However, patients with bone metastasis were mostly diagnosed as end-stage cancer, and their survival status was hard to predict by the suggested model during the acute phase. This few-shot learning model demonstrated potential to fit unmet clinical needs and proved its reliability to extract eigen parameters. We expect that a large sample size and longitudinal follow-ups could facilitate the machine-learning approach to find more significant parameters and prediction model performance. 

## 5. Conclusions

We performed classification based on few-shot learning that provides multiple advantages in analyzing high-dimensional cytokine data with a relatively small sample size. This ML algorithm integrates clinical indices, pathology labeling, and molecular biological aspects as well as assists feature finding from complex parameters. In this study, few-shot learning demonstrated promising accuracy in identifying comprehensive cytokine parameters that had been excluded by conventional stepwise multiple regression analysis. ML suggested a prediction model for the local control of bone metastatic lesions and allowed for the further dissection of the mechanism underlying postablation cytokine patterns during the acute inflammatory phase. A larger metalearning approach that merges cytokine monitoring and immunological observation may improve the long-term local control after MRgFUS. Thus, combined thermal ablation therapy that maintains feasible antitumor inflammation may benefit patients with bone metastases in terms of survival and quality of life.

## Figures and Tables

**Figure 1 cancers-14-00445-f001:**
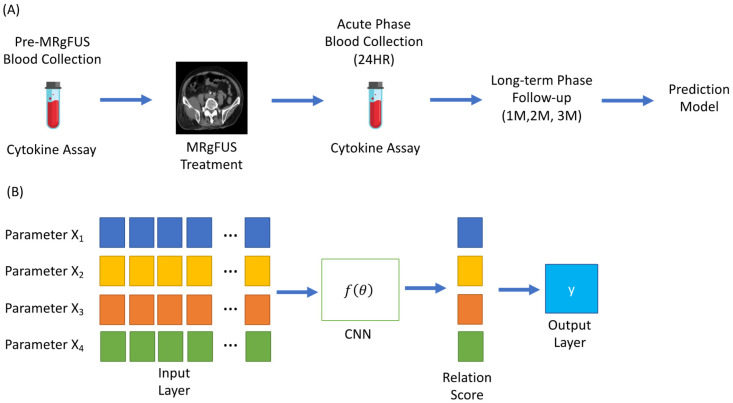
(**A**) Work flow of cytokine parameter collection; (**B**) schematic of few-shot learning approach for parameter classification.

**Figure 2 cancers-14-00445-f002:**
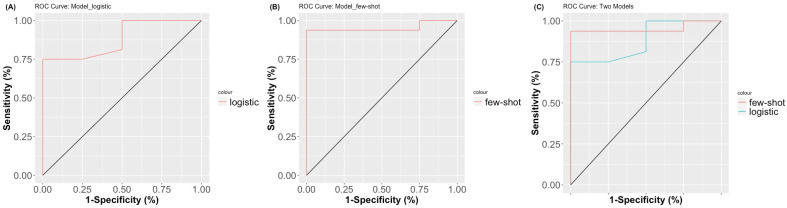
ROC curves of each prediction model. (**A**) Multivariable logistic regression model with stepwise parameter selection (AUC: 0.88); (**B**) few-shot learning classification (AUC: 0.95); (**C**) merged ROC curve.

**Table 1 cancers-14-00445-t001:** Patient status and demographics of overall dataset.

Patient Characteristics	Mean ± SD (Range)
**Number of Patents**	20
**Age, median (range), year**	63.95 ± 10.57 (46–83)
**Gender**	
Male	8 (40%)
Female	12 (60%)
**Pretreatment KPS, median (range)**	83 ± 7.33 (70–90)
Primary Tumor	
Breast cancer	6 (30%)
Lung cancer	8 (40%)
Prostate cancer	1 (5%)
Colon cancer	3 (15%)
Renal cell carcinoma	2 (10%)
**Treated Site**	
Rib	1 (5%)
Sternum	1 (5%)
Acetabulum	1 (5%)
Ilium	3 (15%)
Ischium	1 (5%)
Sacroiliac joint	6 (30%)
Sacrum	5 (25%)
Scapula	2 (10%)
**Pre-treatment Pain Score (NRS)**	
Mean ± std (range)	6.65 ± 1.72 (4–9)
4–6	9 (45%)
7–10	11 (55%)

**Table 2 cancers-14-00445-t002:** Baseline cytokine and post-treatment cytokine status of MRgFUS thermal ablation for bone metastasis.

Cytokine	Pretreatment (Mean ± SD)	Post-Treatment (Mean ± SD)	*p* Value
IL-6	5.54 ± 3.81	3.62 ± 4.38	0.049 *
Exotaxin	27.37 ± 12.91	20.76 ± 12.80	0.067
IL-13	3.50 ± 2.82	2.371 ± 1.45	0.075
IP-10	442.21 ± 394.86	316.28 ± 293.8	0.004 *
IL-1b	1.64 ± 1.43	1.85 ± 2.27	0.514
IL-1ra	86.11 ± 159.23	137.23 ± 369.18	0.313
IL-2	7.31 ± 25.04	17.73 ± 71.01	0.217
IL-4	0.72 ± 0.37	0.72 ± 0.45	0.920
IL-5	6.35 ± 3.57	5.93 ± 3.19	0.200
IL-7	4.41 ± 1.81	4.03 ± 1.48	0.182
IL-8	15.52 ± 45.29	8.21 ± 6.98	0.472
IL-9	10.39 ± 5.15	9.29 ± 4.83	0.188
IL-10	13.96 ± 40.13	32.69 ± 122.5	0.335
IL-12	14.68 ± 19.33	25.56 ± 69.23	0.366
IL-17	12.18 ± 12.16	15.53 ± 26.36	0.468
FGF	17.51 ± 12.98	19.84 ± 25.09	0.506
G-CSF	32.99 ± 60.98	55.98 ± 160.06	0.327
INF-gamma	48.21 ± 33.46	48.94 ± 40.01	0.889
MCP-1	7.01 ± 13.40	7.27 ± 19.96	0.870
MIP-1a	2.223 ± 2.74	2.362 ± 4.36	0.759
MIP-1b	15.13 ± 5.42	13.30 ± 5.73	0.192
PDGF	129.72 ± 128.51	203.48 ± 278.72	0.296
RANTES	1388.5 ± 306.75	1492.9 ± 429.96	0.425
TNF-α	26.96 ± 41.13	27.80 ± 49.54	0.733
VEGF	10.06 ± 11.96	15.26 ± 35.62	0.394

* Paired Mann–Whitney U test significance: *p* value < 0.05; IL, interleukin; FGF, fibroblast growth factor; G-CSF, granulocyte colony-stimulating factor; GM-CSF, granulocyte macrophage colony-stimulating factor; IFN, interferon; IP-10, interferon gamma-induced protein 10; MCP-1, monocyte chemoattractant protein 1; MIP, macrophage inflammatory protein; PDGF, platelet-derived growth factor; RANTES, regulated upon Activation, Normal T cell Expressed, and Secreted; TNF, tumor necrosis factor; VEGF, vascular endothelial growth factor.

**Table 3 cancers-14-00445-t003:** Summary of treatment parameters for MRgFUS.

Treatment Parameters	Mean ± SD
Number of Sonication	21.42 ± 6.21
Duration of treatment (min)	74.38 ± 41.21
Average acoustic power (W)	51.08 ± 19.97
Average energy applied (J)	1029.93 ± 360.46
Temperature (°C)	67.71 ± 7.36

**Table 4 cancers-14-00445-t004:** Univariate and multivariate analyses of parameters for prediction of clinical responses to MRgFUS.

Parameter	Univariate Analysis: Logistic Regression			Multivariable Analyses: Logistic Regression		
	OR	95% CI	*p* Value	OR	95% CI	*p* Value
Age, median (range), years	0.96	0.83–1.07	0.51	1.02	0.83–1.24	0.81
Gender (male vs. female)	0.15	0.01–1.52	0.13			
Pretreatment pain score (NRS)	0.85	0.41–1.66	0.64			
Pretreatment KPS	1.27	1.05–1.74	0.04 *	1.23	1.00–1.74	0.04 *
IL-1b	1.46	0.61–10.39	0.53			
IL-6	0.72	0.48–0.96	0.04 *	0.77	0.48–1.08	0.16
Exotaxin	0.94	0.83–1.03	0.19			
IL-10	1.08	0.98–1.87	0.68			
IL-13	1.13	0.75–2.23	0.62			
IP-10	0.99	0.98–0.99	0.07			
IL-17a	1.06	0.95–1.36	0.43			

* *p* value < 0.05.

**Table 5 cancers-14-00445-t005:** Suggested regression models for analysis of the relationship between clinical characteristics and local control response.

Relationship Formalization	Accuracy	AUC	AIC
Logistic regression: $y\;\sim \;0.2144\cdot X_{KPS}+\left(-0.2676\right)\cdot X_{IL6}+0.02164\cdot X_{Age}$	0.88	0.88	19.35
Few-shot learning model: $\begin{array}{c} y\;\sim \;0.15765\cdot X_{KPS}+\left(-1.058355\right)\cdot X_{IL6}+\left(-0.002651\right)\cdot X_{INFg}\\ +\left(-0.028724\right)\cdot X_{TNFa}+\left(0.33388\right)\cdot X_{VEGF} \end{array}$	0.95	0.95	17.14

## Data Availability

The data presented in this study are available from the corresponding author upon request.

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
