# Peer review of "A Few-Shot Learning Approach Assists in the Prognosis Prediction of Magnetic Resonance-Guided Focused Ultrasound for the Local Control of Bone Metastatic Lesions"

_cancers, 2022, doi:10.3390/cancers14020445_

Round 1
Reviewer 1 Report
This paper describes application of few-shot learning approach for providing a more accurate and precise prognosis model for analysis and prediction of MRgFUS in bone metastatic control. The approach of using machine learning algorithm for integration of clinical indices, pathology labeling, and molecular biology to predict local control and success of bone metastasis ablation is an interesting approach and valuable to share with the community. However, the manuscript requires major English revisions as the current version is difficult to follow.
Comments:
Please revise the English throughout the manuscript. Here are some examples.
Page 2, line 62: Please rephrase the sentence to
It was observed that the circulation of Anti-inflammatory……IP-1 decreased significantly within 24 hours during the acute phase.
Page 2-line 63, please rephrase the sentence.
…..were also observed to increase insignificantly and lead to an anti-inflammatory status….
Line 70, please provide the extended version of AUP
Abstract and discussion section: IL-6 and IP-10 is initially referred to as anti-inflammatory cytokines, but later on discussed as pro-inflammatory cytokines. Please revise to be consistent.
Page 2-line 69, please correct the sentence
The best predictive model included few-shot selected significant cytokines: IL-6, IL-13, 69 IP-10 and Eotaxin which has performed accuracy of 85.2%.....
Line 77: Metastasis is a major complication…..
Line 85: Over 50% of patients…..This sentence needs to be rephrased and corrected grammatically.
Line 94: ….(MRgFUS), which provides a conformal…., Please revise the English and grammers.
Line 98:…..constraint….
Line 103: “MRgFUS provides high resolution” ? What resolution are you referring to? Image? Ultrasound guided treatmetnt?
Line 110:….have been validated to be released in plasma…..
Please describe what GLM, AIC, ROC, and other models and indices are representing and how they are used for model optimization.
Line 210: Continuous variables, such as cytokine concentration….
Table 1: Number of Patients. Please correct spelling. RCC? RCC appears for the first time and abbreviation should be clarified. The table needs to get organized.
Section 2.2 MRgFUS treatment: Please provide the specification of FUS treatment parameters (sonication power, frequency).
Line 235: were recorded within (not with) at least 3 months…..
Line 236: plete? response (CR) whining?! What does whining 3 moths mean? What is plete? Please correct the English.
Line 293: 72% of patients…..
Line 304: correct and rephrase the sentence.
Reviewer 2 Report
This is an interesting paper, with an appropriate methodological approach and adequate design: indeed the modulation of cytokines (as well as other circulating biomarkers) is still not comprehensively investigated and clarified after MRgFUS. The interplay between diseases progression and the short term effect of delivered hyperthermia is an interesting aspect that merits to be further investigation.
The presented work is methodologically sound, even if the reduced number of cases and the inclusion of different targeted lesions render the results less robust. As a consequence, the discussion lacks deep analysis and does not reach a clear assessment with the identification trends and evident correlation.
Also, the conclusion that the crucial factor which determines pro/anti tumorigenesis is a dynamic combination of cytokines, is not sound and clearly supported by evidence. It's reasonable that the complex interplay of different cytokines (together with other signalling proteins) modulates the relapse of the disease.
These aspects need to be comprehensively discussed and the limitation of this study need to be better evidenced and taken into proper account in the frame of a revision.
Since this work is moving towards the development of methods for understanding the prognostic factors of MRgFUS for pain palliation therapy in bone metastases, I believe is of interest to the scientific community after major revision as been indicated.
Round 2
Reviewer 1 Report
The English of the manuscript has been significantly improved. Following are some suggestions for the current version of the manuscripts.
Line 52: MRI imagines replaced by MRI imaging
Line 52: Immune-related Not im-mune
Line 106: deleting the marked section will effect the meaning of the sentence. I recommend not to delete it.
line 170: deleting the marked section will effect the meaning of the sentence. I recommend not to delete it.
line 187: Please correct the word dilu-tions to dilutions
Line 217: Please rephrase the highlighted sentence to passive for consistency.
line 227 and 228: Please rephrase the highlighted sentence to passive for consistency.
line 341: 72% of the patients. Please put "of the"
line 368: 1. change the sentence to passive grammatically. 2. The word best is better to be deleted as the technique has some limitations.
(In this study, a prediction model of few-shot machine learning technique was evaluated
by parameter selection and model performance)
line 392: Please rephrase the sentence to passive grammatically for consistency.
Reviewer 2 Report
The Authors provided the response to the raised question, not fully addressing a critical analysis.
The answer repeated the consideration and the conclusion (i.e., that the crucial factor which determines pro/anti tumorigenesis is a dynamic combination of cytokines) without a critical point of view.
As well as the intrinsic limitation on this work has not been properly discussed.
The proposed model provides a baseline of cytokine in plasma: this is not a limitation, since it's not realistic to measure the cytokine levels in situ in different time points in patients. Other aspects need to be mentioned, e.g., a small number of cases, the unclear inclusion criteria (according to the type and target lesions), the presence of other morbidities, the lacking of a proper validation (by imaging modalities to check the effect in a longitudinal way) of the prognostic value, have not been properly considered and discussed against the relevant literature.
The revision and reply to Author did not modify the shortcomings and lack of a critical view, in particular the quality of the prognostic model, although targeting a good question wort to be investigated.
Round 3
Reviewer 2 Report
The Author has adequately discussed the issues raised and added further information to the manuscript which is now eligible for publication.